# Serum interferon-gamma-induced protein 10 levels can help predict sarcopenia development in patients with primary hepatocellular carcinoma: A retrospective cohort study

Hitomi Takada*, Leona Osawa, Yasuyuki Komiyama, Masaru Muraoka, Yuichiro Suzuki, Mitsuaki Sato, Shoji Kobayashi, Takashi Yoshida, Shinichi Takano, Shinya Maekawa, Nobuyuki Enomoto

Gastroenterology and Hepatology Department of Internal Medicine, Faculty of Medicine, University of Yamanashi, Yamanashi, Japan

* takadahi0107@gmail.com

## Abstract

### Background

Sarcopenia is a prognostic factor in patients with hepatocellular carcinoma (HCC). However, the mechanism underlying sarcopenia development in these patients remains unclear. The chemokine interferon-gamma-induced protein 10/C-X-C motif chemokine ligand 10 (IP-10) has been found to be associated with muscle regeneration or destruction. Thus, we aimed to clarify the role of serum IP-10 levels in predicting sarcopenia development in patients with HCC.

### Methods

This retrospective study enrolled 120 patients with primary HCC whose serum IP-10 levels were measured both at baseline and 1 year after the confirmed diagnosis of HCC. Patients who had sarcopenia at baseline computed tomography imaging were assigned to the Sarco-base group, whereas those in whom sarcopenia was found for the first time after 3 years were assigned to the Sarco-develop group. Those who never met the criteria during the follow-up period were assigned to the Non-Sarco group.

### Results

The baseline IP-10 levels were significantly lower in the Sarco-base group compared to the rest (p = 0.016). Conversely baseline IP-10 levels and IP-10 ratio at 1 year were higher in the Sarco-develop group than in the Non-Sarco group (p = 0.0017, p = 0.025). High IP-10 levels at baseline, and high IP-10 ratios at 1 year were independently related factors for sarcopenia development.

**Data availability statement:** All relevant data are within the manuscript.

**Funding:** The author(s) received no specific funding for this work.

**Competing interests:** The authors have declared that no competing interests exist.

## Conclusions

Patients with sarcopenia at baseline more frequently presented with low IP-10 levels than those without. Contrarily, the group without sarcopenia at baseline and with high baseline IP-10 levels and high IP-10 ratios at 1 year were more likely to develop sarcopenia after 3 years. Monitoring of IP-10 levels may enable the identification of groups prone to develop sarcopenia in patients with HCC.

## Introduction

Sarcopenia is characterized by a low skeletal muscle mass, weakness, and decreased overall physical performance [1]. Based on the evaluation criteria developed by the Japan Society of Hepatology (JSH) for sarcopenia in liver disease cases, patients with chronic liver disease exhibiting low grip strength and muscle mass, as determined by computed tomography (CT) or bioelectrical impedance analysis, are considered to have sarcopenia [2]. Recently, there has been a growing emphasis on evaluating sarcopenia in terms of muscle quality, especially focusing on decreased grip strength or increased intramuscular fat mass [3–6]. Sarcopenia reportedly is a poor prognostic factor in patients with hepatocellular carcinoma (HCC) [7–11]. Thus, sarcopenia has recently gained more attention in those with chronic liver disease.

However, the development of sarcopenia in patients with HCC is often associated with primary sarcopenia due to age and secondary sarcopenia, making it difficult to understand the pathogenesis in some cases. Several theories about sarcopenia development, besides primary sarcopenia, have been proposed; however, the precise mechanism in patients with HCC still requires elucidation. The decreased levels of amino acids and testosterone, hyperammonemia, hypermetabolic state, and suppression of the mammalian target of rapamycin pathway due to lack of mobility and systemic treatment are all thought to contribute to development of sarcopenia [12].

Chemokine interferon-gamma (IFN-γ)-induced protein 10/C-X-C motif chemokine ligand 10 (IP-10), a downstream molecule of IFN-γ, may potentially be involved in the mechanism [13,14]. IP-10 levels were reported to promote the proliferation and differentiation of satellite cells by signaling through the receptor CXCR3. Intramuscular recombinant IP-10 treatment in aged mice induces the proliferation of satellite cells and provokes the increase in the number of regenerated myofibers [15]. Contrarily, the reduction in muscle atrophy via down-regulation of IP-10 levels has been reported in tumor-bearing mice [16]. The serum IP-10 levels correlated with the clinical degree of muscle involvement in patients with systemic sclerosis, suggesting that high IP-10 levels do not usually indicate increased muscle regeneration [17]. The IP-10 levels may produce an opposite effect on muscle regeneration depending on age, presence of chronic disease, presence of cancer, and so on. Therefore, in the present study, we aimed to examine the association between serum IP-10 levels and sarcopenia development in patients with HCC.

## Methods

### Patients

From a total of 738 patients at our hospital with a confirmed diagnosis of primary HCC from January 2008 to January 2021, 239 patients with Barcelona Clinic Liver Cancer (BCLC) stage A who had satisfactory imaging at baseline and were aged ≥20 years were selected, whereas those without sufficient blood samples for IP-10 assay, those with missing data, those diagnosed with HCC other than BCLC stage A, or those with a shorter follow-up (<3 years) were excluded from the analysis. The presence of sarcopenia was further assessed at baseline and after 3 years, and patients were classified into the following three groups; patients with sarcopenia at baseline were classified as the Sarco-base group, patients who met the criteria for sarcopenia after 3 years were classified as the Sarco-develop group and patients who never met the criteria at follow-up were classified as the Non-Sarco group. In each group, 40 patients whose first visit date were closer to the present, i.e., with a short serum sample cryopreservation period were selected, and total 120 patients whose serum IP-10 levels were measured both at baseline and after 1 year, were enrolled in our research. We accessed their data for research purposes on 01/04/2024.

HCC was diagnosed based on the outcomes of the pathological evaluation or the non-rim hyperenhancement in the arterial phase of dynamic CT or gadolinium ethoxybenzyl diethylenetriamine penta-acetic acid-contrast-enhanced magnetic resonance imaging (MRI) and non-peripheral washout or threshold increase, where only nodules that exhibited LR-4 and LR-5 using Liver Imaging Reporting and Data System were categorized as HCC [18].

All patients provided written informed consent prior to study participation, and the study was approved by the Human Ethics Review Committee of Yamanashi University Hospital (approval number: 1326), in accordance with the Declaration of Helsinki.

### Diagnosis of sarcopenia

According to the evaluation criteria for sarcopenia in liver disease established by JSH, the manifestation of decreased grip strength and low muscle mass are indicative of sarcopenia. Nonetheless, due to the retrospective nature of our research, adequate grip strength analysis was not possible. Hence, CT values were used for assessing skeletal muscle quality.

The psoas muscle mass index (PMI) at the level of the third lumbar vertebra, as determined by CT imaging, was applied as an indicator of muscle mass volume. CT images obtained during the primary HCC diagnosis served as the baseline data. The cross-sectional areas of the bilateral psoas muscles were evaluated by manual tracing, and PMI was determined by normalizing these areas to a square of a patient's height in meters. The cut-off value for PMI was set as 6.36 and 3.92 $cm^2/m^2$ for men and women, respectively, as per the JSH criteria [19]. The CT values for the multifidus muscle at the third lumbar vertebral level were indicative of skeletal muscle quality [20–22]. The cut-off value for low CT values was established at 44.4 and 39.3 Hounsfield Unit (HU) for men and women, respectively [3,4]. The measurement of muscle mass volume and CT values was carried out by two hepatology specialists who were experts in this field.

In our research, sarcopenia was outlined as the presence of both low PMI and CT values.

### Evaluation of serum IP-10 levels

Serum samples were gathered from 9 mL of blood obtained at baseline and 1 year after the confirmed diagnosis of HCC. These samples were distributed into aliquots and stored at −80 °C until further analysis. The serum IP-10 levels were measured using 50-µL of the stored serum and an enzyme-linked immunosorbent assay kit, as per the manufacturer's instructions. The levels were determined using standard calibration curves and expressed in pg/mL. The IP-10 ratio was defined as the ratio of IP-10 levels obtained at baseline and 1 year after the confirmed diagnosis of HCC.

## Statistical analysis

All experimental data were expressed as medians (ranges). Between-group comparisons were conducted using the Mann–Whitney U-, Kruskal–Wallis, and Friedman tests along with nonparametric analysis of variance. If the one-way analysis of variance yielded significant results, differences between individual groups were analyzed with Fisher's exact test. A $p < 0.05$ was regarded as statistically significant. All the analyses were done with EZR (Saitama Medical Center, Jichi Medical University, Saitama, Japan), a graphical user interface for R (The R Foundation for Statistical Computing, Vienna, Austria). Specifically, EZR is a modified version of the R commander developed to employ statistical functions commonly applied in biostatistics [23].

## Results

### Patient characteristics

The median age of the patients (n = 120) was 71 years (range: 51–87 years), and 73 patients were male. The Child–Pugh scores were 5, 6, and 7 in 86, 22, and 12 cases, respectively. Regarding the Tumor-Node-Metastasis (TNM) staging, 52 and 68 patients were categorized as having Stage 1 and 2, respectively. Forty patients did not experience sarcopenia (Non-Sarco group), 40 had sarcopenia after 3 years (Sarco-develop group), and 40 patients were assigned to the Sarco-base group. A summary of the features of these patients is shown in Table 1.

### Characteristics of IP-10 levels in BCLC stage A patients

The IP-10 levels only showed a very weak correlation with Albumin-bilirubin (ALBI) score, Child-Pugh score, Branched-chain amino acids (BCAA), Branched-chain amino acids/tyrosine ratio (BTR), platelet counts, alpha-fetoprotein (AFP), and tumor size, with no alternative indicators (Table 2).

### Association between IP-10 levels and sarcopenia development

The baseline IP-10 levels were significantly lower in the Sarco-base group compared to the rest (88 vs. 110 pg/ml, p = 0.016) (Fig 1a). No association was found between the presence of sarcopenia at baseline and IP-10 ratio (0.98 vs. 1.0, p = 0.81) (Fig 1b). Contrarily, IP-10 levels at 1 year after the confirmed diagnosis of HCC were lower in the Non-Sarco group compared to the rest (25 vs. 62 pg/ml, p < 0.001) (Fig 1c). The IP-10 ratio after 1 year was also lower in the Non-Sarco group compared to the rest (0.91 vs. 1.1, p = 0.044) (Fig 1d).

### Comparison of the association between IP-10 levels and sarcopenia development among the three patient groups

Further comparisons were performed between the Sarco-base, Sarco-develop, and Non-Sarco groups. The baseline IP-10 levels were higher in the Sarco-develop group than in the Sarco-base (p < 0.001) and Non-Sarco (p = 0.0017) groups (Fig 2a). The IP-10 ratios were higher in the Sarco-develop group than in the Non-Sarco group (p = 0.025) (Fig 2b). The outcomes of the comparative analysis of the patient characteristics among the three groups are summarized in Table 1.

### Factors associated with sarcopenia development

In all the patients, independently related factors for sarcopenia development were age > 68 years old, male, body mass index <24, with diabetes mellitus, high IP-10 levels at baseline, and high IP-10 ratios at 1 year (Table 3). After propensity score matching using the factors; age, gender, body mass index, and diabetes mellitus, high IP-10 levels at baseline and high IP-10 ratios at 1 year were still independently associated factors for the development of sarcopenia (Table 3).

**Table 1. Baseline characteristics of 120 BCLC stage A patients.**

| | All (n = 120) | Sarco-base group (n = 40) | Sarco-develop group (n = 40) | Non-Sarco group (n = 40) | p value |
|---|---|---|---|---|---|
| Age, years old | 71 (51-87) | 72 (56-83) | 72 (58-84) | 65 (51-87) | 0.017 |
| Male, n | 75 (63%) | 36 (90%) | 24 (60%) | 15 (38%) | 0.017 |
| Body mass index | 23 (15-40) | 22 (17-29) | 22 (18-31) | 26 (15-40) | 0.007 |
| Diabetes mellitus, n | 44 (37%) | 32 (80%) | 28 (70%) | 16 (40%) | 0.001 |
| Habitual drinker, n | 12 (10%) | 2 (5.0%) | 6 (15%) | 4 (10%) | 0.33 |
| Etiology (viral/nonviral), n | 93/27 (78/23%) | 34/6 (85/15%) | 32/8 (80/20%) | 27/13 (67/33%) | 0.16 |
| Child-Pugh grade (A/B), n | 109/11 (91/9%) | 38/2 (95/5%) | 36/4 (90/10%) | 35/5 (87/13%) | 0.20 |
| Albumin-bilirubin grade (1/2a/2b), n | 59/34/27 (49/28/23%) | 24/10/6 (60/25/15%) | 11/14/15 (27/35/38%) | 24/10/6 (60/25/15%) | 0.013 |
| Branched-chain amino acids, μmol/ml | 463 (276-973) | 456 (311-973) | 423 (276-589) | 506 (338-588) | 0.22 |
| Branched-chain amino acids/tyrosine ratio | 5.3 (2.4-11) | 5.0 (2.9-11) | 4.5 (2.4-8.4) | 5.6 (2.5-8.3) | 0.13 |
| Platelet, × $10^3$/μl | 115 (53-258) | 130 (57-258) | 97 (53-176) | 111 (57-211) | 0.026 |
| Alpha-fetoprotein, ng/ml | 4.8 (1.4-6800) | 4.8 (1.4-6800) | 6.5 (172-118) | 4.2 (1.8-945) | 0.42 |
| Des-γ-carboxy prothrombin, mAU/ml | 21 (8.0-7397) | 22 (11-7397) | 28 (8.0-2115) | 18 (12-54) | 0.025 |
| Tumor size, maximum, mm | 19 (8.0-55) | 19 (8.0-45) | 26 (11-35) | 18 (10-55) | 0.14 |
| The number of intrahepatic tumors, n | 1 (1-3) | 1 (1-2) | 1 (1-3) | 1 (1-3) | 0.061 |
| Up to seven beyond at baseline, n | 9 (7.5%) | 0 | 7 (18%) | 2 (5.0%) | 0.009 |
| Therapy for primary HCC (resection/ percutaneous puncture treatment/ transarterial chemoembolization), n | 29/51/40 (24/43/33%) | 18/17/5 (45/43/12%) | 5/10/25 (13/26/61%) | 6/24/10 (15/60/25%) | < 0.001 |
| IP-10 levels at baseline, pg/ml | 100 (36-456) | 88 (40-261) | 171 (39-456) | 91 (36-401) | < 0.001 |
| IP-10 ratio at 1 year | 1.0 (0.16-3.0) | 0.98 (0.16-3.0) | 1.2 (0.45-3.0) | 0.91 (0.21-2.0) | 0.045 |

**Table 2. Factors associated with IP-10 levels in BCLC stage A patients.**

| | Spearman correlation | p value |
|---|---|---|
| Age, | 0.11 | 0.25 |
| Body mass index | -0.056 | 0.54 |
| PMI | -0.15 | 0.10 |
| CT value | -0.12 | 0.19 |
| Child-Pugh score | 0.30 | < 0.001 |
| Albumin-bilirubin score | 0.46 | < 0.001 |
| Branched-chain amino acids, μmol/ml | -0.24 | 0.021 |
| Branched-chain amino acids/tyrosine ratio | -0.30 | 0.0043 |
| Platelet, × $10^3$/μl | -0.26 | 0.0041 |
| Alpha-fetoprotein | 0.32 | < 0.001 |
| Des-γ-carboxy prothrombin | 0.094 | 0.31 |
| Tumor size, maximum | 0.26 | 0.0049 |
| The number of intrahepatic tumors | 0.17 | 0.067 |

## Characteristics of patients with high baseline IP-10 levels or high ratios

Patients without sarcopenia at baseline but who had high IP-10 levels at baseline or high IP-10 ratios after 1 year exhibited several distinguishing characteristics, which were as follows: high ALBI grade, low BCAA, low BTR, low platelet counts, more patients opting for transarterial chemoembolization (TACE) as treatment for primary HCC, and recurrence beyond up to seven criteria during the 3-year follow-up period (Table 4).

 

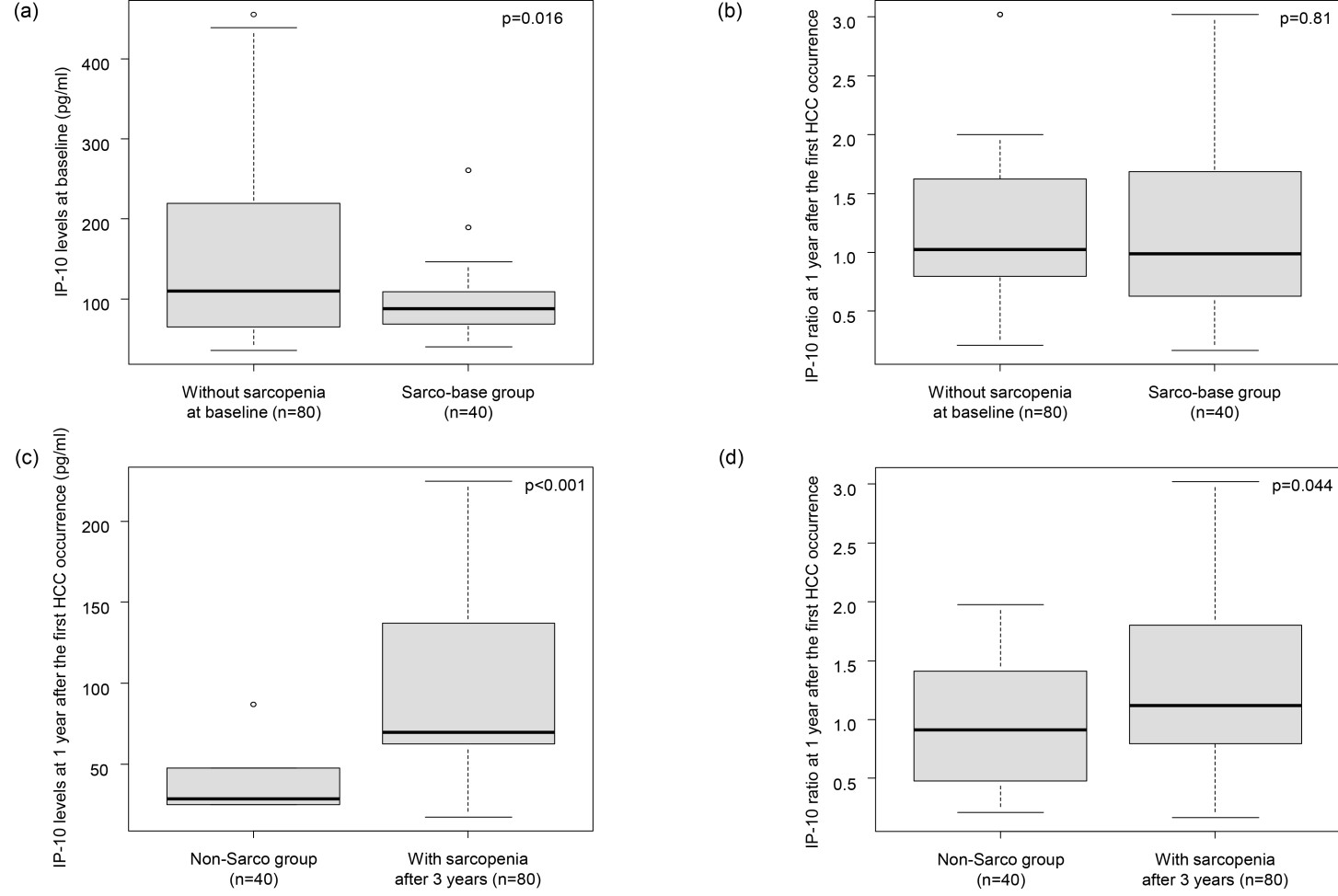

**Fig 1. Association between IP-10 levels and sarcopenia development.** (a) The association between serum IP-10 levels and the presence of sarcopenia at baseline, (b) The association between IP-10 ratios and the presence of sarcopenia at baseline. (c) The association between serum IP-10 levels at 1 year after the first HCC occurrence and sarcopenia development. (d) The association between IP-10 ratios at 1 year after the first HCC occurrence and sarcopenia development.

## Discussion

In this study, we analyzed the relationship between sarcopenia and serum IP-10 levels in patients with BCLC stage A HCC. Our findings demonstrated that patients with sarcopenia at baseline had lower IP-10 levels than those without. Contrarily, in patients without sarcopenia at baseline, sarcopenia was more likely to develop in those with higher baseline IP-10 levels and higher 1-year IP-10 ratios. This indicates that high IP-10 levels have an opposite effect on muscle regeneration, depending on the timing after the confirmed diagnosis of HCC. To the best of our knowledge, our results clarified for the first time the relationship between sarcopenia and IP-10 levels in patients with primary HCC, and our data may aid in the identification of those patients at a high-risk of developing sarcopenia and contribute to the development of therapeutic intervention methods.

The term sarcopenia was first introduced in 1989, and since then, numerous studies have analyzed the correlation between sarcopenia and the prognosis of patients with HCC [8,9,11,24]. Nonetheless, the molecular mechanisms serving as a basis of the development of sarcopenia in patients with HCC are complex and have not been completely understood.

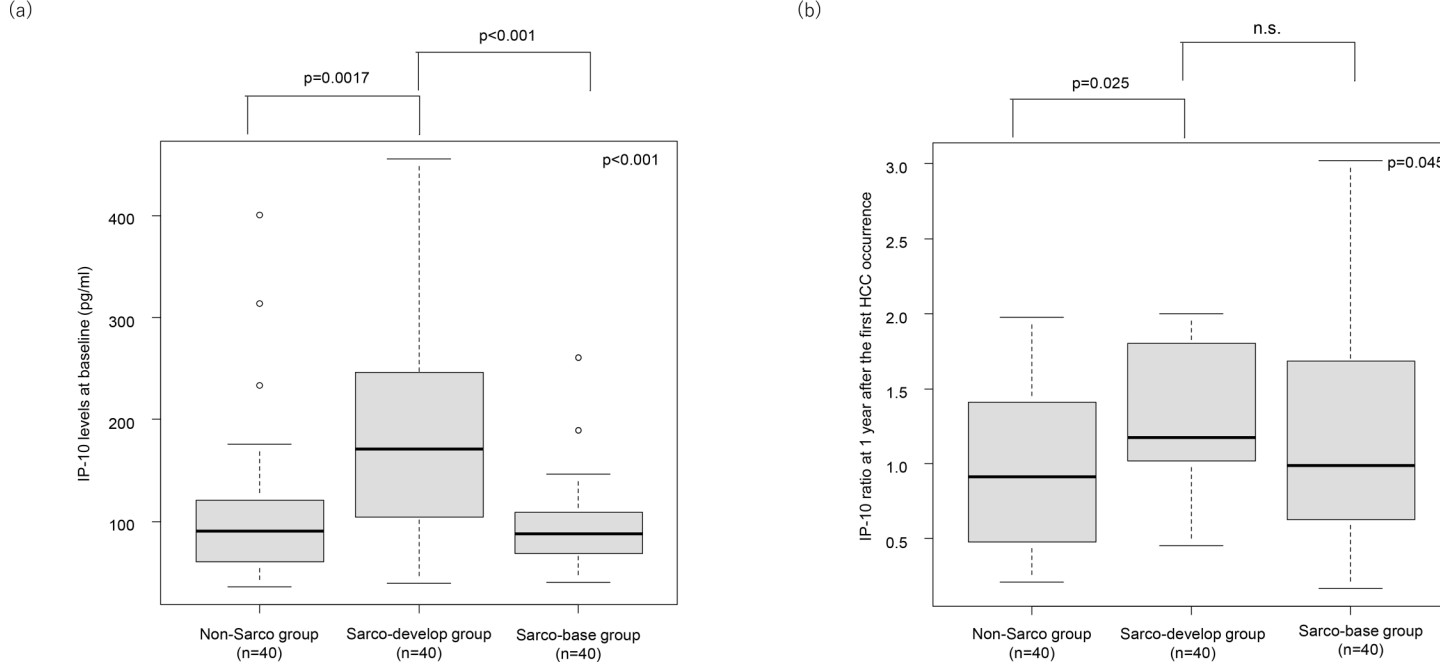

**Fig 2. Association between IP-10 levels and sarcopenia development among the three patient groups.** (a) Baseline IP-10 levels among three groups. (b) IP-10 ratios among three groups.

Particularly, the relationship between IP-10 levels and sarcopenia development in patients with HCC has not been thoroughly investigated.

Previous papers have reported higher peripheral IP-10 concentrations in the elderly than in the younger patients [25]. Aging is also associated with a chronic state of increased plasma levels of pro-inflammatory mediators, such as tumor necrosis factor α (TNFα), interleukin 6 (IL-6) and C-reactive protein (CRP), hence IP-10 may reflect levels in the inflammatory environment [26]. A decrease in muscle atrophy and tumor shrinkage via down-regulation of the IP-10 levels have been demonstrated in tumor-bearing mice [16]. IFN-γ may facilitate muscle damage by inhibiting M2 macrophage activation and muscle cell proliferation in a mouse model of muscular dystrophy [27]. Serum IP-10 levels were also shown to be correlated with the severity of muscle involvement in patients with systemic sclerosis [17]. Conversely, gene expression patterns associated with the responses to IFN-γ was considerably downregulated during muscle regeneration in aged mice, which is thought to promote satellite cell dysfunctions in aged skeletal muscles [15]. Intramuscular recombinant IP-10 treatment in aged mice provoked the proliferation of satellite cells and caused an increase in regenerated myofibers [15]. IP-10 activated myogenic differentiation in vitro, indicative of its possible direct influence on muscle regeneration [28]. Thus, it has been proposed that IP-10 levels may have an opposing effect on muscle regeneration, and that the effects may also vary according to the presence of malignancies and chronic diseases.

Indeed, the interpretation of the relationship between IP-10 levels and sarcopenia in patients with HCC poses substantial difficulties. Previous studies have indicated an association between the IP-10 levels and cancer stage, with higher levels detected in patients with advanced HCC. The overexpression of IP-10 has been linked to serum AFP levels, tumor size and number along with TNM stage [29]. Patients with higher IP-10 expression levels also had considerably lower overall and disease-free survival rates [30]. The downregulation of IP-10 can suppress metastasis and tumor invasion in HCC patients [28]. Deletion of IP-10 in fibrosis-associated HCC mice leads to the enhancement of anti-tumoral immune cells and an overall reduction of chemokines, but to a specific accumulation of chemokines at the tumor site [31,32]. Thus,

**Table 3. Factors associated with sarcopenia development in BCLC stage A patients.**

| | Before matching | | | | After matching | | | |
|---|---|---|---|---|---|---|---|---|
| | Univariate analysis | | Multivariate analysis | | Univariate analysis | | Multivariate analysis | |
| | Odds ratio | p value | Odds ratio | p value | Odds ratio | p value | Odds ratio | p value |
| Age, years old | 1.1 (1.01-1.1) | 0.011 | | | 0.99 (0.93-1.1) | 0.80 | | |
| > 68 | 3.3 (1.4-6.8) | 0.0052 | 8.9 (2.3-36) | 0.0015 | | | | |
| Male | 5.0 (2.2-11) | < 0.001 | 44 (12-73) | < 0.001 | 0.85 (0.28-2.6) | 0.78 | | |
| Body mass index | 0.87 (0.79-0.95) | 0.0023 | | | 0.90 (0.79-1.04) | 0.16 | | |
| < 24 | 2.9 (1.3-6.5) | 0.0076 | 3.9 (1.1-13) | 0.030 | | | | |
| With diabetes mellitus, | 4.5 (2.0-10) | < 0.001 | 22 (4.6-102) | < 0.001 | 0.85 (0.28-2.6) | 0.78 | | |
| Habitual drinker | 0.37 (0.077-1.8) | 0.21 | | | 1.7 (0.41-6.8) | 0.48 | | |
| Etiology, viral | 2.3 (0.94-5.5) | 0.067 | | | 5.4 (0.92-29) | 0.068 | | |
| Child-Pugh grade B | 0.55 (0.15-2.0) | 0.36 | | | 3.2 (0.26-39) | 0.37 | | |
| Albumin-bilirubin grade 2a/2b | 1.7 (0.67-4.1) | 0.28 | | | 3.7 (0.36-5.9) | 0.58 | | |
| Branched-chain amino acids, μmol/ml | 0.99 (0.99-1.0) | 0.40 | | | 0.99 (0.99-1.01) | 0.17 | | |
| Branched-chain amino acids/tyrosine ratio | 0.90 (0.70-1.2) | 0.39 | | | 0.69 (0.47-1.03) | 0.068 | | |
| Platelet, × $10^3$/μl | 1.0 (0.99-1.01) | 0.96 | | | 0.99 (0.97-1.01) | 0.056 | | |
| Alpha-fetoprotein, ng/ml | 1.0 (0.99-1.0) | 0.47 | | | 1.01 (0.99-1.02) | 0.25 | | |
| Des-γ-carboxy prothrombin, mAU/ml | 1.02 (0.99-1.1) | 0.27 | | | 1.03 (0.99-1.1) | 0.17 | | |
| Tumor size, maximum, mm | 0.98 (0.95-1.01) | 0.21 | | | 0.96 (0.91-1.01) | 0.091 | | |
| The number of intrahepatic tumors, n | 0.98 (0.71-1.4) | 0.91 | | | 1.1 (0.74-1.7) | 0.60 | | |
| Up to seven beyond at baseline | 1.8 (0.36-9.2) | 0.47 | | | 3.6 (0.66-20) | 0.14 | | |
| IP-10 levels at baseline, pg/ml | 1.01 (1.01-1.1) | 0.042 | | | 1.01 (1.0-1.02) | 0.022 | | |
| > 100 | 2.2 (1.1-4.7) | 0.045 | 3.3 (1.03-11) | 0.045 | 2.0 (1.3-12) | 0.046 | 3.5 (1.3-15) | 0.048 |
| IP-10 ratio at 1 year | 2.0 (1.1-3.8) | 0.026 | | | 9.8 (2.1-46) | 0.0036 | | |
| > 1.02 | 2.8 (1.3-6.1) | 0.011 | 8.1 (2.2-30) | 0.0015 | 18 (3.0-108) | 0.0016 | 14 (3.3-59) | <0.001 |

IP-10 levels have also been reported to modulate the tumor microenvironment of HCC. In this study, more patients with high baseline IP-10 levels or high 1-year ratios showed recurrence beyond up to seven criteria during the follow-up period. IP-10 levels may be associated not only with the development of sarcopenia but also with tumor modulation.

Furthermore, the IP-10 levels reportedly are associated with liver function, disease progression, and cirrhosis. Understanding the IP-10 dynamics in patients with chronic hepatitis C may be useful for predicting liver function after direct-acting antiviral therapy [33]. In patients with chronic hepatitis B, the IP-10 levels increased in the group with declining liver function [34]. The IP-10 levels in patients with non-alcoholic steatohepatitis were higher than in those in the control group and patients with non-alcoholic fatty liver [35]. Herein, the IP-10 levels were also found to be correlated with ALBI and

**Table 4. Characteristics of patients presenting high IP-10 levels at baseline or high IP-10 ratios, without sarcopenia at baseline.**

| | The rest of patients (n = 22) | Patients with high baseline IP-10 levels or high 1-year ratios (n = 58) | p value |
|---|---|---|---|
| Age, years old | 67 (51-75) | 71 (52-87) | 0.58 |
| Male, n | 12 (55%) | 27 (47%) | 0.62 |
| Body mass index | 24 (15-31) | 24 (18-40) | 0.39 |
| Diabetes mellitus, n | 12 (55%) | 22 (41%) | 0.32 |
| Habitual drinker, n | 0 | 10 (17%) | 0.054 |
| Etiology (viral/nonviral), n | 16 (73%) | 43 (74%) | 1.0 |
| Child-Pugh grade (A/B), n | 22/0 (100/0%) | 49/9 (84/16%) | 0.069 |
| Albumin-bilirubin grade (1/2a/2b), n | 16/6/0 (73/27/0%) | 19/18/21 (33/31/36%) | < 0.001 |
| Branched-chain amino acids, μmol/ml | 506 (423-568) | 444 (276-589) | 0.020 |
| Branched-chain amino acids/tyrosine ratio | 5.8 (3.3-6.6) | 4.9 (2.4-8.4) | 0.008 |
| Platelet, × $10^3$/μl | 127 (84-206) | 94 (53-211) | 0.015 |
| Alpha-fetoprotein, ng/ml | 4.2 (1.8-93) | 6.3 (1.7-945) | 0.10 |
| Des-γ-carboxy prothrombin, mAU/ml | 18 (12-49) | 21 (8.0-2115) | 0.41 |
| Tumor size, maximum, mm | 25 (13-55) | 18 (10-48) | 0.060 |
| The number of intrahepatic tumors, n | 1 (1-3) | 1 (1-3) | 0.20 |
| Up to seven beyond at baseline, n | 0 | 9 (16%) | 0.057 |
| Therapy for primary HCC (resection/ percutaneous puncture treatment/ transarterial chemoembolization (TACE)), n | 14/8/0 (64/36/0%) | 9/19/30 (15/33/52%) | < 0.001 |
| Recurrence during the 3-year follow-up period, n | 7 (32%) | 29 (50%) | 0.21 |
| Up to seven beyond recurrence during the 3-year follow-up period, n | 0 | 17 (29%) | 0.004 |
| TACE performed more than 3 times during the 3-year follow-up period, n | 4 (18%) | 22 (38%) | 0.11 |

Child–Pugh scores, and high IP-10 levels at baseline or IP-10 ratios were associated with high ALBI grade, low BCAA, low BTR, low platelet count, more patients opting for TACE as treatment for primary HCC, and recurrence beyond up to seven criteria during the follow-up period. These results indicate that high IP-10 levels may be associated with advanced cirrhosis or HCC. Sarcopenia is associated with low grade systemic inflammation, as indicated by increased inflammatory cytokines, leading to oxidative stress. Moreover, inflammation and stress-related signaling pathways are important in the progression of fibrosis and HCC development, and may have been associated with the present results [36,37]. IP-10 levels in HCC patients are associated with inflammation and tumor formation in liver tissue, suggesting that the association with the development of sarcopenia may be more robust than in patients with other diseases. However, as the analysis by etiology was insufficient in this study, future analysis involving a larger number of cases is needed.

The present study delved into the baseline IP-10 levels and their changes as a potential molecular mechanism underlying sarcopenia development. The low baseline IP-10 levels may indicate that the baseline sarcopenia was due to the decreased levels of IP-10, a factor facilitating muscle differentiation, as described above, or it may reflect muscular atrophy due to sarcopenia, causing an inability to secrete IP-10, presumably a myokine, which is produced by skeletal muscles [14,15,28]. Contrarily, the intramuscular CD4 T-cells are increased in both young and aged mice during a viral infection, whereas the number of intramuscular CD8 T-cells increased only in aged muscle [38]. Similar mechanisms may be at work in patients with advanced cirrhosis or poorly controlled HCC. Moreover, it has been proposed that an increasing percentage of CXCR3 variants over time might be an essential component of endometriosis-related carcinogenesis in patients with ovarian cancer [39]. Our findings, taken together with previous reports, may suggest that high IP-10 ratios in BCLC stage A patients may be associated with the development of sarcopenia via dysregulation of T-cell transfer to

muscle, T-cell differentiation and receptor abnormalities [40]. Further validation of this assumption, e.g., by measuring T-cell subsets, is needed in the future. In addition, in this study, among the group with high baseline IP-10 levels or high 1-year ratios, patients who had undergone TACE more than twice during the follow-up period developed sarcopenia more frequently than those who had undergone TACE less than twice (86 vs. 50%, p = 0.006). This suggests that disease progression and therapeutic interventions may influence the mechanism by which IP-10 influences muscle mass.

The present study has some limitations. First, it had a single-center retrospective design and lacked grip strength measurements. Thus, integrating grip strength evaluations, required by the currently widely used guidelines for sarcopenia diagnosis in Japan, is a challenge in future. Second, patient selection bias may have been present. The baseline IP-10 levels and the trends remained independent factors for the development of sarcopenia after matching. In contrast, the Non-Sarco group in the analysis of all the patients was younger, and had fewer thin, diabetic and male subjects, a bias that influenced the results of this study. Additional analyses in a large number of patients are needed on the reality of this situation and the importance of IP-10 levels using matching. Third, the relationship between etiology and IP-10 levels has not been fully analyzed. The study should be re-evaluated in the future with a larger number of cases. Moreover, IP-10 levels are an item influenced by many factors and it is premature to relate the association between the development of sarcopenia and IP-10 levels in this study alone. Further prospective studies of this association, including other risk factors for the development of sarcopenia, are warranted.

The present study is the first to demonstrate that the IP-10 levels at baseline or IP-10 ratios may be associated with the development of sarcopenia in patients with BCLC stage A HCC. By evaluating the IP-10 levels, this study not only provides data for identifying high-risk groups for sarcopenia development but also improves our knowledge on the underlying mechanisms involved in sarcopenia development in patients with primary HCC.

## Conclusion

Patients with sarcopenia at baseline were more likely to present with low IP-10 levels than those without. Conversely, those not presenting with sarcopenia at the first occurrence of HCC and subsequently developed sarcopenia after 3 years had higher baseline IP-10 levels and higher IP-10 ratios at 1 year. This indicates that the IP-10 levels may have different effects on the muscles depending on the timing after the confirmed diagnosis of HCC. Furthermore, monitoring the IP-10 levels may be an effective tool for defining high-risk groups in patients with primary HCC, with a predisposition to develop sarcopenia, which is a significant prognostic factor for patients with primary HCC.

## Acknowledgments

We thank Ms. Takako Ohmori and Ms. Tomoko Nakajima for their valuable technical assistance.

## Author contributions

**Conceptualization:** Hitomi Takada.

**Data curation:** Hitomi Takada, Leona Osawa, Yasuyuki Komiyama, Masaru Muraoka, Yuichiro Suzuki, Mitsuaki Sato, Shoji Kobayashi, Takashi Yoshida, Shinichi Takano, Shinya Maekawa.

**Formal analysis:** Hitomi Takada.

**Methodology:** Hitomi Takada.

**Supervision:** Shinya Maekawa, Nobuyuki Enomoto.

**Validation:** Hitomi Takada.

**Visualization:** Hitomi Takada.

**Writing – original draft:** Hitomi Takada.

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
