## [Decision Letter · Decision Letter 0]

9 Oct 2024

PONE-D-24-35270Serum interferon-gamma-induced protein 10 levels can help predict sarcopenia development in patients with primary hepatocellular carcinoma: A retrospective cohort studyPLOS ONE

Dear Dr. Takada,

Thank you for submitting your manuscript to PLOS ONE. After careful consideration, we feel that it has merit but does not fully meet PLOS ONE’s publication criteria as it currently stands. Therefore, we invite you to submit a revised version of the manuscript that addresses the points raised during the review process.

**Please specifically address the comments made by reviewer #1. **

We look forward to receiving your revised manuscript.

Kind regards,

Amir Radfar, MD,MPH,MSc,DHSc

Academic Editor

PLOS ONE

Reviewers' comments:

Reviewer's Responses to Questions

**Comments to the Author**

1. Is the manuscript technically sound, and do the data support the conclusions?

Reviewer #1: Yes

Reviewer #2: Yes

2. Has the statistical analysis been performed appropriately and rigorously? 

Reviewer #1: Yes

Reviewer #2: Yes

3. Have the authors made all data underlying the findings in their manuscript fully available?

Reviewer #1: No

Reviewer #2: Yes

4. Is the manuscript presented in an intelligible fashion and written in standard English?

Reviewer #1: Yes

Reviewer #2: Yes

5. Review Comments to the Author

Reviewer #1: I would like to thank the authors for the privilege of reading their research. Whether IP10 levels in the serum can be used to predict the extent of sarcopenia in primary hepatocellular carcinoma and in extrapolation the prognosis in this patient group is a novel question. The introduction or discussion could possibly add more context to the role of IP10 in inflammation and tumorigenesis [1].

As there is existing research that suggests that IP10 modulates the tumour microenvironment of fibrosis associated HCC [2] [3] , as well as higher levels of inflammatory markers in age-related sarcopenia [4], I would be interested to know the association between disease progression with IP10 levels and sarcopenia in this context as I wonder whether the levels of IP10 actually reflect the levels of the pro-inflammatory environment and affect tumour modulation as well as the development of sarcopenia.

The authors have referred to some breadth of literature regarding how IP10 levels are correlated with sarcopenia and I am curious whether the strength of this correlation is different in people with HCC compared to other groups of patients exhibiting sarcopenia. Is there an aetiological inflammation and sarcopenia as well as inflammation and tumorigenesis on the liver tissue and does IP10 have a role in both these pathways? I understand if the discussion may be outside the scope of the paper, however I wonder whether the authors think this finding has shown a role for future research into this central question.

In the methods section it is unclear to me what the original clinical indication was to measure the IP10 levels of the study group initially - was this routine, random or was it measured in a specific subgroup of patients? The authors acknowledge that there may be some selection bias and I believe answering this question will help the audience understand the extent to which such bias may exist. I would also like to know how these 120 people equally distributed in between the three groups were then selected from the base cohort. Was there any specific randomisation algorithm or was it entirely up to the researcher's judgement?

It is interesting to note that the opposing effects of IP10 on muscle regeneration seen in previous studies has been encountered again and that too in the same patient population with different effects on muscle mass at diagnosis and at follow up after treatment has been initiated. Does this imply that the mechanism by which IP10 influences muscle mass changes with disease progression or with treatment.

I note that the master table with the anonymised data collected has not been provided. However the data has been presented in a summarised form with the median and range of different attributes being presented.

I would like to conclude that in my humble opinion, this is an engaging research article and with its limitations that the authors have acknowledged, lends more insight into the mechanisms underlying the systemic effects of hepatocellular carcinoma.

References.

1. Sasya Madhurantakam, Zachary J Lee, Aliya Naqvi, Shalini Prasad, Importance of IP-10 as a biomarker of host immune response: Critical perspective as a target for biosensing, Current Research in Biotechnology, Volume 5, 2023, 100130, ISSN 2590-2628, https://doi.org/10.1016/j.crbiot.2023.100130.

2. Brandt EF, Baues M, Wirtz TH, May JN, Fischer P, Beckers A, Schüre BC, Sahin H, Trautwein C, Lammers T, Berres ML. Chemokine CXCL10 Modulates the Tumor Microenvironment of Fibrosis-Associated Hepatocellular Carcinoma. Int J Mol Sci. 2022 Jul 23;23(15):8112. doi: 10.3390/ijms23158112. PMID: 35897689; PMCID: PMC9329882.

3. Chen S, Zhang L, Chen Y, Zhang X, Ma Y. Chronic Inflammatory and Immune Microenvironment Promote Hepatocellular Carcinoma Evolution. J Inflamm Res. 2023;16:5287-5298 https://doi.org/10.2147/JIR.S435316

4. Dalle S, Rossmeislova L, Koppo K. The Role of Inflammation in Age-Related Sarcopenia. Front Physiol. 2017 Dec 12;8:1045. doi: 10.3389/fphys.2017.01045. PMID: 29311975; PMCID: PMC5733049

Reviewer #2: Thank you for the opportunity for reviewing the manuscript “Serum interferon-gamma-induced protein 10 levels can help predict sarcopenia development in patients with primary hepatocellular carcinoma: A retrospective cohort study”.

This is an interesting manuscript, which should be deemed an exploratory one. The authors studied the association of IP-10 with sarcopenia in patients with hepatocellular carcinoma. However, IP-10 values are influenced by many factors, as is recognized. So, although the statistical association exists in this group of patients, it seems too early to adopt the IP-10 level for intervention in individual patients. It is essential to study this association prospectively and include other factor which should be considered important as potential prognostic factors for sarcopenia.

6. PLOS authors have the option to publish the peer review history of their article (what does this mean? ). If published, this will include your full peer review and any attached files.

**Do you want your identity to be public for this peer review?** For information about this choice, including consent withdrawal, please see our Privacy Policy .

Reviewer #1: No

Reviewer #2: No

---

## [Author Response · Author response to Decision Letter 1]

22 Nov 2024

Dear Dr. Amir Radfar,

Academic Editor

and the reviewers, PLOS ONE

PONE-D-24-35270

Serum interferon-gamma-induced protein 10 levels can help predict sarcopenia development in patients with primary hepatocellular carcinoma: A retrospective cohort study

Thank you for your kind review of our manuscript.

We appreciate receiving this opportunity to re-submit our manuscript.

Considering the suggestions, we have written Point by Points. We believe that these changes will alleviate the concerns of the reviewers. We would greatly appreciate if our manuscript could be accepted for publication in PLOS ONE.

Sincerely,

Hitomi Takada, MD.

Gastroenterology and Hepatology Department of Internal Medicine, Faculty of Medicine,

University of Yamanashi,

Yamanashi, Japan

Point by points

Reviewer 1

1. The introduction or discussion could possibly add more context to the role of IP10 in inflammation and tumorigenesis [1]. As there is existing research that suggests that IP10 modulates the tumour microenvironment of fibrosis associated HCC [2] [3] , as well as higher levels of inflammatory markers in age-related sarcopenia [4], I would be interested to know the association between disease progression with IP10 levels and sarcopenia in this context as I wonder whether the levels of IP10 actually reflect the levels of the pro-inflammatory environment and affect tumour modulation as well as the development of sarcopenia.

Thank you very much for your detailed comments. As you pointed out, the relationship between IP-10 levels, levels of the pro-inflammatory environment and tumor modulation was not adequately described in our manuscript.

Thank you also for the important references.

We have modified the text as follows.

(line 235, page 21)

Aging is also associated with a chronic state of increased plasma levels of pro-inflammatory mediators, such as tumor necrosis factor α (TNFα), interleukin 6 (IL-6) and C-reactive protein (CRP), hence IP-10 may reflect levels in the inflammatory environment [26].

(line 257, page 22)

Deletion of IP-10 in fibrosis-associated HCC mice leads to the enhancement of anti-tumoral immune cells and an overall reduction of chemokines, but to a specific accumulation of chemokines at the tumor site [31, 32]. Thus, IP-10 levels have also been reported to modulate the tumor microenvironment of fibrosis-associated HCC. In this study, more patients with high baseline IP-10 levels or 1-year ratios showed recurrence beyond up to seven criteria during the follow-up period. IP-10 levels may be associated not only with the development of sarcopenia but also with tumor modulation.

2. The authors have referred to some breadth of literature regarding how IP10 levels are correlated with sarcopenia and I am curious whether the strength of this correlation is different in people with HCC compared to other groups of patients exhibiting sarcopenia. Is there an aetiological inflammation and sarcopenia as well as inflammation and tumorigenesis on the liver tissue and does IP10 have a role in both these pathways? I understand if the discussion may be outside the scope of the paper, however I wonder whether the authors think this finding has shown a role for future research into this central question.

Thank you very much for your comments. Reasons suggesting that the association between IP-10 levels, sarcopenia and tumorigenesis may be stronger than in patients with other diseases are that systemic inflammation and oxidative stress-related signaling pathways are important not only in the development of sarcopenia, but also in fibrosis progression and HCC.

We have modified the text as follows.

(line 275, page 23)

Sarcopenia is associated with low grade systemic inflammation, as indicated by increased inflammatory cytokines, leading to oxidative stress. Moreover, inflammation and stress-related signaling pathways are important in the progression of fibrosis and HCC development, and may have been associated with the present results [36, 37]. IP-10 levels in HCC patients are associated with inflammation and tumor formation in liver tissue, suggesting that the association with the development of sarcopenia may be more robust than in patients with other diseases.

3. In the methods section it is unclear to me what the original clinical indication was to measure the IP10 levels of the study group initially - was this routine, random or was it measured in a specific subgroup of patients? The authors acknowledge that there may be some selection bias and I believe answering this question will help the audience understand the extent to which such bias may exist. I would also like to know how these 120 people equally distributed in between the three groups were then selected from the base cohort. Was there any specific randomisation algorithm or was it entirely up to the researcher's judgement?

Thank you very much for your comments. As you pointed out, our description was inappropriate.

We have modified the text as follows.

(line 83, page 5)

From a total of 738 patients at our hospital with a confirmed diagnosis of primary HCC from January 2008 to January 2021, 239 patients with Barcelona Clinic Liver Cancer (BCLC) stage A who had satisfactory imaging at baseline and were aged ≥20 years were selected, whereas those without sufficient blood samples for IP-10 assay, those with missing data, those diagnosed with HCC other than BCLC stage A, or those with a shorter follow-up (<3 years) were excluded from the analysis. The presence of sarcopenia was further assessed at baseline and after 3 years, and patients were classified into the following three groups; patients with sarcopenia at baseline were classified as the Sarco-base group, patients who met the criteria for sarcopenia for the first time at 3-year follow-up were classified as the Sarco-develop group and patients who never met the criteria at follow-up were classified as the Non-Sarco group. In each group, 40 patients were selected from those with a newer date of first visit, and total 120 patients whose serum IP-10 levels were measured both at baseline and after 1 year, were enrolled in our research.

4. It is interesting to note that the opposing effects of IP10 on muscle regeneration seen in previous studies has been encountered again and that too in the same patient population with different effects on muscle mass at diagnosis and at follow up after treatment has been initiated. Does this imply that the mechanism by which IP10 influences muscle mass changes with disease progression or with treatment.

Thank you very much for your detailed comments. As you pointed out, the possibility that disease progression and treatment interventions may alter the relationship between IP-10 levels and the development of sarcopenia is an interesting point.

We have modified the text as follows.

(line 296, page 24)

In addition, in this study, among the group with high baseline IP-10 levels or 1-year ratios, patients who had undergone TACE more than twice during the follow-up period developed sarcopenia more frequently than those who had undergone TACE less than twice (86 vs. 50%, p=0.006). This suggests that disease progression and therapeutic interventions may influence the mechanism by which IP-10 influences muscle mass.

Reviewer 2

1. IP-10 values are influenced by many factors, as is recognized. So, although the statistical association exists in this group of patients, it seems too early to adopt the IP-10 level for intervention in individual patients. It is essential to study this association prospectively and include other factor which should be considered important as potential prognostic factors for sarcopenia.

Thank you very much for your detailed comments. As you pointed out, it is premature to relate the association between the development of sarcopenia and IP-10 levels in this study alone. Prospective studies including other risk factors for the development of sarcopenia, are warranted in future.

(line 308, page 24)

Moreover, IP-10 levels are an item influenced by many factors and it is premature to relate the association between the development of sarcopenia and IP-10 levels in this study alone. Further prospective studies of this association, including other risk factors for the development of sarcopenia, are warranted.

---

## [Decision Letter · Decision Letter 1]

7 Jan 2025

PONE-D-24-35270R1Serum interferon-gamma-induced protein 10 levels can help predict sarcopenia development in patients with primary hepatocellular carcinoma: A retrospective cohort studyPLOS ONE

Dear Dr. Takada,

Thank you for submitting your manuscript to PLOS ONE. After careful consideration, we feel that it has merit but still does not fully meet PLOS ONE’s publication criteria as it currently stands. Therefore, we invite you to submit a revised version of the manuscript that addresses the points raised during the review process.

Especially:

1-Please provide a detailed point-by-point response to the reviewers in the re-submission package, explaining how each concern was addressed.2-Please include precise details on patient inclusion and exclusion criteria, clarifying the rationale for the chosen patient groups and the term 'newer date of first visit'.3-Please propose a revised table structure that reduces redundancy while ensuring critical data is not omitted.4-Please clarify which data support the statement on T-cell dysregulation. If this is based on indirect evidence, acknowledge the limitation explicitly. If additional data (e.g., T-cell subsets, cytokine levels) are available, include them to substantiate the argument.5- Please include detailed multivariable analysis and provide an interpretation of how confounders impact the results.

We look forward to receiving your revised manuscript.

Kind regards,

Amir Radfar, MD,MPH,MSc,DHSc

Academic Editor

PLOS ONE

Reviewers' comments:

Reviewer's Responses to Questions

**Comments to the Author**

1. If the authors have adequately addressed your comments raised in a previous round of review and you feel that this manuscript is now acceptable for publication, you may indicate that here to bypass the “Comments to the Author” section, enter your conflict of interest statement in the “Confidential to Editor” section, and submit your "Accept" recommendation.

Reviewer #2: All comments have been addressed

Reviewer #3: (No Response)

Reviewer #4: (No Response)

2. Is the manuscript technically sound, and do the data support the conclusions?

Reviewer #2: Yes

Reviewer #3: Yes

Reviewer #4: (No Response)

3. Has the statistical analysis been performed appropriately and rigorously? 

Reviewer #2: Yes

Reviewer #3: I Don't Know

Reviewer #4: (No Response)

4. Have the authors made all data underlying the findings in their manuscript fully available?

Reviewer #2: Yes

Reviewer #3: Yes

Reviewer #4: (No Response)

5. Is the manuscript presented in an intelligible fashion and written in standard English?

Reviewer #2: Yes

Reviewer #3: Yes

Reviewer #4: (No Response)

6. Review Comments to the Author

Reviewer #2: As stated above, my comments were satisfactorily responded to, I have no more remarks for the authors. In my view, the manuscript may be published as it is.

Reviewer #3: Takada and colleagues explored the role of IFN-gamma induced protein 10 (IP-10) levels in patients with HCC and its role in the development of sarcopenia. In a retrospective study of 120 patients’ authors compared IP-10 levels between patient with baseline sarcopenia, patients with sarcopenia after 3 years and patients who never developed sarcopenia measured at baseline and 1y after diagnosis. Results of the analysis showed significantly higher baseline IP-10 levels in patients whom developed sarcopenia at 3-years as compared to patients with baseline sarcopenia or whom never developed sarcopenia. They also show that IP-10 level at 1 year were lower in patient who never developed sarcopenia in comparison with the rest of the patient. They add some additional analysis with IP-10 ratios as further evidence. They conclude that dynamics in IP-10 levels could be used to identify patients with HCC at risk for developing sarcopenia, while correctly acknowledging that correlation of IP-10 levels with sarcopenia does not necessarily imply causality of sarcopenia (low IP-10 levels cause or consequence of sarcopenia).

The authors present a well written manuscript in standard English with little need for additional proofreading. The article flows in a natural way but the reader can get caught up with the numerous tables presented and some ambiguities.

My remarks:

- In the context of development of sarcopenia authors use “at 3 years” and “during 3 years” interchangeable throught the script. Doing the analysis “at 3 years” would minimize immortal time bias. Examples: line 33, 91, 148, …

- For these finding to be extrapolated to other populations patient selection has to be described in a more transparent fashion (line 92-95). “Newer date of first visit” has to be specified. Why exactly (these) 40 patients in each group?

- There are 5 tables presenting aggregated patient data. Could some tables be joined together? Also, in Table 4 IP-10 levels at 1y next to baseline levels could be informative.

- In the discussion section the authors make a statement of an association of IP-10 levels and dysregulation of T-cell response. What result from your analysis support this statement? Are there any further data not provided (measurements of T-cell subsets, cytokines,)?

Reviewer #4: The authors intend to conduct a retrospective cohort study to examine the association between serum IP-10 levels and sarcopenia development in patients with HCC. This study recruited 120 patients with measurements at baseline and 1 year after the diagnosis of HCC. Three groups (Sarco-base, Sarco-develop, and Non-Sarco group) were considered. The authors concluded that monitoring of IP-10 levels may enable the identification of groups prone to develop sarcopenia in patients with HCC.

1. Abstract. The results sound confusing. “baseline IP-10 … significantly lower in the Sarco-base group compared to the rest”, but it further states that “the baseline IP-10 levels were higher in the Sarco-develop group than in the Sarco-base…”.

2. Table 4. I assume that the comparison for Table 4 was conducted with baseline data. But please clarify this.

3. It’s interesting to observe lower IP-10 levels at baseline for patients with sarcopenia but a higher chance of developing sarcopenia for those with higher IP-10 levels at baseline. As all the analyses were conducted between two variables without considering other covariates, was this opposite direction scenario due to a lack of consideration of other confounding variables when quantifying the relationship between IP-10 level and the status of sarcopenia? Multiple variable analysis should be conducted to clarify this.

7. PLOS authors have the option to publish the peer review history of their article (what does this mean? ). If published, this will include your full peer review and any attached files.

**Do you want your identity to be public for this peer review?** For information about this choice, including consent withdrawal, please see our Privacy Policy .

Reviewer #2: No

Reviewer #3: No

Reviewer #4: No

---

## [Author Response · Author response to Decision Letter 2]

16 Feb 2025

Dear Dr. Amir Radfar,

Academic Editor

and the reviewers, PLOS ONE

PONE-D-24-35270

Serum interferon-gamma-induced protein 10 levels can help predict sarcopenia development in patients with primary hepatocellular carcinoma: A retrospective cohort study

Thank you for your kind review of our manuscript.

We appreciate receiving this opportunity to re-submit our manuscript.

Considering the suggestions, we have written Point by Points. We believe that these changes will alleviate the concerns of the reviewers. We would greatly appreciate if our manuscript could be accepted for publication in PLOS ONE.

Sincerely,

Hitomi Takada, MD.

Gastroenterology and Hepatology Department of Internal Medicine, Faculty of Medicine,

University of Yamanashi,

Yamanashi, Japan

Point by points

Editor

1- Please provide a detailed point-by-point response to the reviewers in the re-submission package, explaining how each concern was addressed.

Thank you for your kind guidance.

We will describe the amendments in the form of point by point.

2- Please include precise details on patient inclusion and exclusion criteria, clarifying the rationale for the chosen patient groups and the term 'newer date of first visit'.

Thank you very much for your detailed comments. As you pointed out, the details on patient inclusion and exclusion criteria were not fully explained in the original manuscript. Moreover, the term ‘newer date of first visit’ was used to refer to patients whose first visit date was closer to the present, i.e. with a short serum sample cryopreservation period. However, the statement was not stated with enough consideration. We have modified the text as follows.

(line 91, page 4)

In each group, 40 patients whose first visit date were closer to the present, i.e. with a short serum sample cryopreservation period were selected, and total 120 patients whose serum IP-10 levels were measured both at baseline and after 1 year, were enrolled in our research.

3- Please propose a revised table structure that reduces redundancy while ensuring critical data is not omitted.

Thank you very much for your detailed comments. As you pointed out, the tables in the original manuscript were redundant and difficult to read. We have merged Tables 1 and 4, deleted Table 3 and revised them for easier readability. Moreover, we have corrected the statement regarding the table numbers in the text.

(line 183, page 12)

The outcomes of the comparative analysis of the patient characteristics among the three groups are summarized in Table 1.

(line 201, page 16)

Patients without sarcopenia at baseline but who had high IP-10 levels at baseline or IP-10 ratios after 1 year exhibited several distinguishing characteristics, which were as follows: high ALBI grade, low BCAA, low BTR, low platelet counts, more patients opting for TACE as treatment for primary HCC, and recurrence beyond up to seven criteria during the 3-year follow-up period (Table 4).

4- Please clarify which data support the statement on T-cell dysregulation. If this is based on indirect evidence, acknowledge the limitation explicitly. If additional data (e.g., T-cell subsets, cytokine levels) are available, include them to substantiate the argument.

Thank you very much for your detailed comments. As you pointed out, we have described an association of IP-10 levels and dysregulation of T-cell response. However, as this is an assumption based on indirect evidence and not the results obtained in this study, the description has been amended as follows.

(line 287, page 22)

Our findings, taken together with previous reports, may suggest that high IP-10 ratios in BCLC stage A patients may be associated with the development of sarcopenia via dysregulation of T-cell transfer to muscle, T-cell differentiation and receptor abnormalities [40]. Further validation of this assumption, e.g. by measuring T-cell subsets, is needed in the future.

5- Please include detailed multivariable analysis and provide an interpretation of how confounders impact the results.

Thank you very much for your detailed comments. As you point out, we should have presented the results of multivariate analysis on sarcopenia development. We have added Table.3, and modified the text as follows.

(line 192, page 13)

Independently related factors for sarcopenia development were age >68 years old, male, body mass index <24, with diabetes mellitus, high IP-10 levels at baseline, and high IP-10 ratios at 1 year (Table 3).

Reviewer #2

Thank you for your kind comments.

Reviewer #3

1- In the context of development of sarcopenia authors use “at 3 years” and “during 3 years” interchangeable throught the script. Doing the analysis “at 3 years” would minimize immortal time bias. Examples: line 33, 91, 148, …

Thank you for your kind guidance. As you pointed out, our description was inappropriate. This study only included cases that had been followed for more than three years, and the presence of sarcopenia development was determined by imaging after three years, which we have believed to some extent to have taken a considered approach to immortality bias. We have modified the text as follows.

(line 31, page 2)

Patients who had sarcopenia at baseline computed tomography imaging were assigned to the Sarco-base group, whereas those in whom sarcopenia was found for the first time after 3 years were assigned to the Sarco-develop group.

(line 42, page 2)

Contrarily, the group without sarcopenia at baseline and with high baseline IP-10 levels and IP-10 ratios at 1 year were more likely to develop sarcopenia after 3 years.

(line 87, page 4)

The presence of sarcopenia was further assessed at baseline and after 3 years, and patients were classified into the following three groups; patients with sarcopenia at baseline were classified as the Sarco-base group, patients who met the criteria for sarcopenia after 3 years were classified as the Sarco-develop group and patients who never met the criteria at follow-up were classified as the Non-Sarco group.

(line 146, page 6)

Forty patients did not experience sarcopenia (Non-Sarco group), 40 had sarcopenia after 3 years (Sarco-develop group), and 40 patients were assigned to the Sarco-base group.

(line 316, page 23)

Conversely, those not presenting with sarcopenia at the first occurrence of HCC and subsequently developed sarcopenia after 3 years had higher baseline IP-10 levels and IP-10 ratios at 1 year.

2- For these finding to be extrapolated to other populations patient selection has to be described in a more transparent fashion (line 92-95). “Newer date of first visit” has to be specified. Why exactly (these) 40 patients in each group?

Thank you very much for your detailed comments. As you pointed out, our description about patient selection was inadequate in the original manuscript. Moreover, the term ‘newer date of first visit’ was used to refer to patients whose first visit date was closer to the present, i.e. with a short serum sample cryopreservation period. However, the statement was not stated with enough consideration. We have modified the text as follows.

(line 91, page 4)

In each group, 40 patients whose first visit date were closer to the present, i.e. with a short serum sample cryopreservation period were selected, and total 120 patients whose serum IP-10 levels were measured both at baseline and after 1 year, were enrolled in our research.

3- There are 5 tables presenting aggregated patient data. Could some tables be joined together? Also, in Table 4 IP-10 levels at 1y next to baseline levels could be informative.

Thank you very much for your detailed comments. As you pointed out, the tables in the original manuscript were redundant and difficult to read. We have merged Tables 1 and 4, deleted Table 3 and revised them for easier readability. We have modified Table 1 (Table 4 in the original manuscript), and the statement regarding the table numbers in the text.

(line 183, page 12)

The outcomes of the comparative analysis of the patient characteristics among the three groups are summarized in Table 1.

(line 201, page 16)

Patients without sarcopenia at baseline but who had high IP-10 levels at baseline or IP-10 ratios after 1 year exhibited several distinguishing characteristics, which were as follows: high ALBI grade, low BCAA, low BTR, low platelet counts, more patients opting for TACE as treatment for primary HCC, and recurrence beyond up to seven criteria during the 3-year follow-up period (Table 4).

4- In the discussion section the authors make a statement of an association of IP-10 levels and dysregulation of T-cell response. What result from your analysis support this statement? Are there any further data not provided (measurements of T-cell subsets, cytokines,)?

Thank you very much for your detailed comments. As you pointed out, we have described an association of IP-10 levels and dysregulation of T-cell response. However, as this is an assumption based on indirect evidence and not the results obtained in this study, the description has been amended as follows.

(line 287, page 22)

Our findings, taken together with previous reports, may suggest that high IP-10 ratios in BCLC stage A patients may be associated with the development of sarcopenia via dysregulation of T-cell transfer to muscle, T-cell differentiation and receptor abnormalities [40]. Further validation of this assumption, e.g. by measuring T-cell subsets, is needed in the future.

Reviewer #4

1- Abstract. The results sound confusing. “baseline IP-10 … significantly lower in the Sarco-base group compared to the rest”, but it further states that “the baseline IP-10 levels were higher in the Sarco-develop group than in the Sarco-base…”.

Thank you very much for your detailed comments. As you pointed out, our description in abstract section was inadequate in the original manuscript. We have modified the text as follows.

(line 37, page 2)

Conversely baseline IP-10 levels and IP-10 ratio at 1 year were higher in the Sarco-develop group than in the Non-Sarco group (p = 0.0017, p = 0.025).

High IP-10 levels at baseline, and high IP-10 ratios at 1 year were independently related factors for sarcopenia development.

2- Table 4. I assume that the comparison for Table 4 was conducted with baseline data. But please clarify this.

Thank you very much for your detailed comments. As you pointed out, comparisons in Table 4 are made with baseline data. Characteristics of the three groups have been modified to present with baseline data in all cases in Table 1.

3- It’s interesting to observe lower IP-10 levels at baseline for patients with sarcopenia but a higher chance of developing sarcopenia for those with higher IP-10 levels at baseline. As all the analyses were conducted between two variables without considering other covariates, was this opposite direction scenario due to a lack of consideration of other confounding variables when quantifying the relationship between IP-10 level and the status of sarcopenia? Multiple variable analysis should be conducted to clarify this.

Thank you very much for your detailed comments. As you point out, we should have presented the results of multivariate analysis on sarcopenia development. We have added Table.3, and modified the text as follows.

(line 192, page 13)

Independently related factors for sarcopenia development were age >68 years old, male, body mass index <24, with diabetes mellitus, high IP-10 levels at baseline, and high IP-10 ratios at 1 year (Table 3).

---

## [Decision Letter · Decision Letter 2]

12 Mar 2025

PONE-D-24-35270R2Serum interferon-gamma-induced protein 10 levels can help predict sarcopenia development in patients with primary hepatocellular carcinoma: A retrospective cohort studyPLOS ONE

Dear Dr. Takada,

Thank you very much for thoroughly addressing the previous comments and significantly improving your manuscript. After careful consideration, we feel that it has merit but does not fully meet PLOS ONE’s publication criteria as it currently stands. Therefore, we invite you to submit a revised version of the manuscript that addresses the points raised during the review process.

Specifically ,

Please address the following points highlighted by Reviewer #5:

**Define Abbreviations:** Clearly define all abbreviations (e.g., ALBI, BCAA, BTR, BCLC) upon their first appearance in the manuscript. This will ensure clarity for readers who may not be familiar with these terms.**Clarify IP-10 Ratios:** Explicitly state in your manuscript how you defined "high" or "low" IP-10 ratios or levels, and provide clinical rationale or references supporting your chosen thresholds.**Discuss Potential Group Biases:** Reviewer #5 observed  demographic differences between your study groups:The Non-Sarco group was younger (median age 65), had a higher BMI (>24), higher diabetes prevalence (>40%), and fewer male subjects compared to the other groups.Since your manuscript identifies age >68, BMI <24, diabetes, and male gender as independently related factors for sarcopenia, these differences could represent potential biases influencing your results. Please explicitly discuss:How these demographic differences could influence your findings.Steps you've taken (or propose to take) to control for or minimize these biases.

Addressing these points clearly and comprehensively will enhance your manuscript’s scientific rigor, accuracy, and credibility.

We look forward to receiving your revised manuscript.

Kind regards,

Amir Radfar, MD,MPH,MSc,DHSc

Academic Editor

PLOS ONE

Reviewers' comments:

Reviewer's Responses to Questions

**Comments to the Author**

1. If the authors have adequately addressed your comments raised in a previous round of review and you feel that this manuscript is now acceptable for publication, you may indicate that here to bypass the “Comments to the Author” section, enter your conflict of interest statement in the “Confidential to Editor” section, and submit your "Accept" recommendation.

Reviewer #4: (No Response)

Reviewer #5: (No Response)

2. Is the manuscript technically sound, and do the data support the conclusions?

Reviewer #4: (No Response)

Reviewer #5: Yes

3. Has the statistical analysis been performed appropriately and rigorously? 

Reviewer #4: (No Response)

Reviewer #5: Yes

4. Have the authors made all data underlying the findings in their manuscript fully available?

Reviewer #4: (No Response)

Reviewer #5: Yes

5. Is the manuscript presented in an intelligible fashion and written in standard English?

Reviewer #4: (No Response)

Reviewer #5: Yes

6. Review Comments to the Author

Reviewer #4: (No Response)

Reviewer #5: Dear Authors,

Thank you for submitting interesting research highlighting the need for better understanding and screening of sarcopenia.

I have few suggestions:

1. Please describe all abbreviation as they appear in manuscript (e.g. ALBI, BCAA, BTR, BCLC) to be more understanadblke for readers not familiar with them

2. Several times (lines 208, 220) you mentioned "high IP-10 value and IP-10 ratio" - please ensure is it high or low ratio level in order to be no doubt for the readers

3. Out of three groups equally divided by numbers which is interesting and not common in research, the Non-sarco group was the "youngest" one 65 Age Median, with BMI >24, and DM in >40% and with less male subjects than in other groups. Would you consider this as bias? Please explain and comment since you stated that age >68, BMI >24 and DM are independently related factor for sarcopenia development

7. PLOS authors have the option to publish the peer review history of their article (what does this mean? ). If published, this will include your full peer review and any attached files.

**Do you want your identity to be public for this peer review?** For information about this choice, including consent withdrawal, please see our Privacy Policy .

Reviewer #4: No

Reviewer #5: No

---

## [Author Response · Author response to Decision Letter 3]

22 Mar 2025

Several theories about sarcopenia development, besides primary sarcopenia, have been proposed; however, the precise mechanism in patients with HCC still requires elucidation.

---

## [Decision Letter · Decision Letter 3]

20 Apr 2025

Serum interferon-gamma-induced protein 10 levels can help predict sarcopenia development in patients with primary hepatocellular carcinoma: A retrospective cohort study

PONE-D-24-35270R3

Dear Dr. Takada,

We’re pleased to inform you that your manuscript has been judged scientifically suitable for publication and will be formally accepted for publication once it meets all outstanding technical requirements.

Kind regards,

Amir Radfar, MD,MPH,MSc,DHSc

Academic Editor

PLOS ONE

Additional Editor Comments (optional):

Reviewers' comments:

Reviewer's Responses to Questions

**Comments to the Author**

1. If the authors have adequately addressed your comments raised in a previous round of review and you feel that this manuscript is now acceptable for publication, you may indicate that here to bypass the “Comments to the Author” section, enter your conflict of interest statement in the “Confidential to Editor” section, and submit your "Accept" recommendation.

Reviewer #5: All comments have been addressed

Reviewer #6: All comments have been addressed

2. Is the manuscript technically sound, and do the data support the conclusions?

Reviewer #5: Yes

Reviewer #6: Yes

3. Has the statistical analysis been performed appropriately and rigorously? 

Reviewer #5: Yes

Reviewer #6: I Don't Know

4. Have the authors made all data underlying the findings in their manuscript fully available?

Reviewer #5: Yes

Reviewer #6: Yes

5. Is the manuscript presented in an intelligible fashion and written in standard English?

Reviewer #5: Yes

Reviewer #6: Yes

6. Review Comments to the Author

Reviewer #5: (No Response)

Reviewer #6: I congratulate authors for this articulated manuscript which will add to the body of knowledge on sarcopenia.

7. PLOS authors have the option to publish the peer review history of their article (what does this mean? ). If published, this will include your full peer review and any attached files.

**Do you want your identity to be public for this peer review?** For information about this choice, including consent withdrawal, please see our Privacy Policy .

Reviewer #5: No

Reviewer #6: No

---

## [Editor Report · Acceptance letter]

PONE-D-24-35270R3

PLOS ONE

Dear Dr. Takada,

I'm pleased to inform you that your manuscript has been deemed suitable for publication in PLOS ONE. Congratulations! Your manuscript is now being handed over to our production team.

Kind regards,

on behalf of

Dr. Amir Radfar

Academic Editor

PLOS ONE